# Public acceptance of privacy-encroaching policies to address the COVID-19 pandemic in the United Kingdom

Stephan Lewandowsky[1,2]*, Simon Dennis[3], Andrew Perfors[3], Yoshihisa Kashima[3], Joshua P. White[3], Paul Garrett[3], Daniel R. Little[3], Muhsin Yesilada[1]

1 University of Bristol, Bristol, United Kingdom, 2 University of Western Australia, Perth, Australia, 3 University of Melbourne, Melbourne, Australia

* stephan.lewandowsky@bristol.ac.uk

**Data Availability Statement:** The data are available at https://osf.io/42wj6/ (first wave) and https://osf.io/pw5yj/(second wave).

## Abstract

The nature of the COVID-19 pandemic may require governments to use privacy-encroaching technologies to help contain its spread. One technology involves co-location tracking through mobile Wi-Fi, GPS, and Bluetooth to permit health agencies to monitor people's contact with each other, thereby triggering targeted social-distancing when a person turns out to be infected. The effectiveness of tracking relies on the willingness of the population to support such privacy encroaching measures. We report the results of two large surveys in the United Kingdom, conducted during the peak of the pandemic, that probe people's attitudes towards various tracking technologies. The results show that by and large there is widespread acceptance for co-location tracking. Acceptance increases when the measures are explicitly time-limited and come with opt-out clauses or other assurances of privacy. Another possible future technology to control the pandemic involves "immunity passports", which could be issued to people who carry antibodies for the COVID-19 virus, potentially implying that they are immune and therefore unable to spread the virus to other people. Immunity passports have been considered as a potential future step to manage the pandemic. We probe people's attitudes towards immunity passports and find considerable support overall, although around 20% of the public strongly oppose passports.

## Introduction

The COVID-19 pandemic has changed nearly all aspects of people's lives around the world. In the absence of a vaccine or successful treatments, the only tools available to control the pandemic are behavioral in nature [1]. Countries that have successfully "flattened the curve" have primarily resorted to social-distancing measures, such as encouraging or forcing people to stay home, restricting public or even private gatherings, restricting movement through public spaces, cancelling large public events, and so on.

However, social distancing cannot be sustained indefinitely, which raises the question about how social life can resume without reigniting the pandemic. This question has become

**Funding:** This work was supported by the Elizabeth Blackwell Institute, University of Bristol, with funding from the University's alumni and friends. The first author was supported by a Humboldt Award from the Humboldt Foundation in Germany during part of this work.

**Competing interests:** The authors have declared that no competing interests exist.

particularly acute at the time of this writing (November 2020), as countries around the world are struggling to manage a second wave of the pandemic after seemingly bringing the virus under control during the boreal summer. At least two potential technological or biomedical options have been developed or proposed to assist with management of the relaxation of social distancing. The first option involves the use of tracking technologies, which monitor people's interactions and send alerts to people who have been in proximity to others who turn out to be infected [2]. This technology has matured to the point where several countries, among them Singapore, Germany, and Australia, rolled out tracking technologies at scale during the boreal summer, with other countries including the U.K. following suit more recently. The second option remains impractical at present and involves issuing people with "immunity passports" if they test positive for antibodies, indicating their presumed immunity to the virus. Immunity passports would bestow privileges on their bearer, such as exemption from social distancing measures.

This article reports the results from two large-scale surveys conducted in the United Kingdom during the height of the first wave of the pandemic (March-April 2020) that probed the public's attitude towards both of these privacy-encroaching options to combat the pandemic: tracking technologies and immunity passports. The principal objective of the survey was to understand which aspects of tracking policies are considered acceptable, and which might be opposed because of their implications for privacy. A second objective of the survey was to identify potential predictors of policy acceptance from a set of candidate attitude constructs, such as political worldviews, trust in government, and perceived risk from COVID.

## Tracking technologies

Several tracking "apps" exist for use in smartphones. A general property of all apps is that they keep track of a person's contacts with others, and if a person turns out to be infected, everyone they encountered during the previous critical time period are identified and alerted via text message. The apps differ, however, in where and how they store the contact information. Some tracking apps involve central storage (e.g., on a government server), whereas others keep all information local and communicate to contacts without that information being knowable by health authorities or the government.

At the policy level in the United Kingdom, multiple developments and analyses have ultimately led to the release of a decentralized tracking app in September 2020, the *NHS COVID-19* app. Although the Coronavirus Bill [3] passed on 25 March 2020 did not include any provisions for wider surveillance tracing, public health enforcement in the U.K. has had existing widespread power to request contact data for infectious or potentially infectious persons.

As early as March 2020, the Information Commissioners' Office (ICO) opined that use of mobile phone data would be legal [4] if broader contact-tracing were introduced (presumably by legislation). The ICO also acknowledged that anonymous geolocation data is already being used to fight the pandemic and approved its use [5].

Setting aside legal considerations, there is no doubt that all existing tracking technologies come with wide-ranging implications for people's privacy. The implications of "public-health surveillance" have stimulated much concern among some scholars [6]. In the U.K., more than 170 cybersecurity and privacy experts signed an open letter in April 2020 [7], warning the government against use of a centralized tracking app for mass surveillance. (The first author of this article was a signatory of that letter.) At the time, the U.K. government was eschewing a decentralized approach that had been jointly developed by Apple and Google. Two months later, the British government discontinued the centralized approach and switched to the Apple/Google Bluetooth-based approach that is at the heart of the *NHS COVID-19* app.

Notwithstanding the shift to a decentralized system, the number of downloads has fallen short of the target required for effective control of the pandemic. As of late October 2020, only half as many people have downloaded the app as needed to effectively halt the spread of the virus [8]. One reason for the insufficient number of downloads may be the public's concerns about privacy. In most societies, people are known to place considerable value on their privacy. A survey of the public in 27 countries within the European Union found that 87% of the public found protection of their privacy to be important or very important [9]. Similarly, in a more recent survey in Germany, 82% of respondents claimed that they are very or at least somewhat concerned about their data privacy [10]. One might therefore expect the public to be concerned about the invasion of privacy that is a nearly inevitable by-product of any COVID-related tracking technology. What is unknown, however, is how specific features of candidate technologies affect people's attitudes, what safeguards might reduce privacy concerns, and how political views and risk perceptions determine views on privacy-encroaching measures.

There are several reasons why privacy concerns might ultimately take a back seat in the context of controlling COVID-19. First, people in the European Union generally endorse the reuse of health data for the common good, although they are concerned about the exploitation of that data through commercialization [11]. Second, there is evidence that people's concerns about privacy are highly context dependent [12]. Specifically, it has been proposed that people engage in a "privacy calculus", such that people will self-disclose personal information, for example by using social media, so long as the perceived benefits exceed the perceived negative consequences [13]. If people engage in a privacy calculus, the variables driving that calculus must be examined and understood.

In the context of mobile apps, recent research has illustrated aspects of users' privacy calculus. For example, one recent study showed that users' privacy concerns were a function of several variables, such as prior privacy experience (in particular violations of privacy), anxiety about the role of computers and automation generally, perceived control over personal data, and concerns about giving permission for an app to use personal data [14]. Although all those variables significantly contributed to privacy concerns, the largest share of variance was explained by the permission concern; that is, reluctance to accept an app's requests to grant access to personal data. Other research has explored the specifics of privacy permissions [15]. This study showed that providing justification for permission requests (e.g., explaining who would have access to the data generated by the app) significantly reduced privacy concerns whereas perceived sensitivity of the requested information enhanced privacy concerns. Moreover, perceived popularity of an app reduced privacy concerns and was directly linked to greater download intentions.

These precedents suggest that users' decisions about app download and usage are attuned to relevant variables. The present context is, however, uniquely different from other apps because tracking technologies do not deliver a direct benefit to users—on the contrary, being notified of a contact with an infected person may benefit public health but the personal consequences, such as self-isolation, are largely negative. Another difference to conventional apps is that contact tracing works best if a large number of people are using the app. The privacy calculus of contact-tracing apps is thus likely to involve a different set of variables, such as perceived risk from COVID-19, that are not part of the typical repertoire of research on app usage and privacy.

## Immunity passports

Although at present immunity passports do not exist, they are within reach as serological tests are becoming available, although to date their reliability has been insufficient [16]. The ethical

implications of immunity passports are hotly debated in the literature [17–20]. The primary concern involves the implications of immunity, which free the person from being subjected to social distancing measures because they are presumed not to be infectious themselves. Conversely, people who are not immune may be confined to their homes and locked out of society [18].

Historical precedent from the 19th century suggests that this division of society into those who are immune and those who are not can have dystopian consequences—affecting all aspects of life, from choice of job to choice of romantic partners [18]. In addition to stratifying along a new dimension of biologically-determined "haves" and "have nots", the mere existence of immunity passports would also trigger an erosion of privacy because passports can only be useful to the extent that their holders are monitored and checked. Passports may also ironically create a risk to public health: If the privileges associated with immunity are sufficiently great, there may be sufficient incentive for people to seek self-infection with the virus [18]. Some scholars have therefore argued that "this idea has so many flaws that it is hard to know where to begin" [18].

On the other side of the ledger, some scholars have argued that under certain circumstances, immunity passports may be ethical, provided sufficient safeguards are put in place [16, 17, 19, 20]. For example, it has been argued that certification of one's immunity status may spur people into greater prosocial altruism, for example by taking on riskier treatment roles or donating blood [17]. Another suggestion has been to prioritize critical or high-risk sectors of society (e.g., health care workers) for testing and to issue immunity passports only to people within that sector, thus limiting inequities to people whose welfare is deemed to be of particular interest to society overall [16].

## The present study

We now report the results from two large-scale surveys conducted in the United Kingdom during the height of the pandemic (March-April 2020) that probed the public's attitude towards both privacy-encroaching options to combat the pandemic: tracking technologies and immunity passports. The surveys presented people with one of several different hypothetical scenarios that described a tracking app, accompanied by different policy options (e.g., a sunset clause for data retention). We also collected a variety of attitude measures, such as people's worldviews, trust in government, and their risk perception relating to COVID, to identify potential predictors of policy acceptance.

## Method

### Overview

The two survey waves were conducted roughly three weeks apart and were nearly identical, with differences noted below. The preregistration for the first survey wave can be found at https://osf.io/d3pcn. The second wave inherited the same preregistration. The surveys reported here are part of a larger, international project that involved data collection in 7 countries (U.K., Australia, U.S.A., Germany, Taiwan, Spain, and Switzerland). A continually-updated summary of the overall project is available at https://stephanlewandowsky.github.io/UKsocialLicence/index.html.

The first wave included two policy scenarios, each revolving around a different hypothetical tracking app: In one scenario, the public had to opt in voluntarily and could choose whether or not do download the app (called the "mild" scenario from here on). In the other scenario, all mobile users were mandated to download the app and the government could issue

quarantine orders and use the tracking data to locate and fine people who were violating those orders ("severe" scenario).

The second wave additionally included a third hypothetical scenario ("Bluetooth"), in which people's phones exchanged messages anonymously whenever they were in proximity, thus alerting people who may have been infected without the government knowing who they are or where they were. Use of this app was voluntary.

The choice of those specific scenarios was based on discussion of apps that resembled or instantiated those scenarios in the media and by government. However, at the time of the surveys, no app was available in the U.K., with limited testing only commencing on 22 April. After a switch in technology, a decentralized app relying on the Google/Apple Bluetooth technology became available for download by the public on 24 September 2020.

## Participants

The first survey was conducted on 28 and 29 March 2020 and involved a representative sample of 2,000 U.K. participants, recruited through the online platform Prolific (https://www.prolific.co/). Prolific stratifies representative samples for age, sex and ethnicity. Participants were at least 18 years old and were paid 85 Pence for their participation in the 10-minute study. At the time, there were 14,543 confirmed cases of COVID-19 in the U.K., with 1,161 deaths.

The second wave was conducted on 16 April and involved another Prolific representative sample of 1,500 participants. Participants were paid GBP 1.34 for their participation in the (approximately) 15-minute study. This was equivalent to GBP 5.98 per hour based on the average observed completion time. At the time of the second wave, there were 98,476 confirmed cases of COVID-19 in the U.K., with 14,915 deaths attributed to the disease. Notwithstanding the relatively brief temporal window between waves, deaths increased 13-fold within less than 3 weeks.

## Instrument and procedure

Verbatim copies of the surveys are available at https://osf.io/d3pcn (for the first wave) and https://osf.io/pw5yj/ (for the second wave). Fig 1 provides an overview of the survey instrument used in both waves. Each white box represents a block with one or more questions pertaining to that topic or construct. Numbers next to the white boxes indicate the number of items in that block. The black boxes represent the different tracking scenarios being tested. Comprehension questions immediately after the scenario and a free text box at the very end of survey (for additional comments) are not shown. The comprehension questions asked participants to state what the scenario they had just read was about. The correct answer ("The government considering using tracking technology to help reduce the spread of COVID-19") had to be selected from among foils such as "using new technology to eliminate influenza" or "developing a vaccine to immunize the population from COVID-19." Respondents who failed the comprehension question were excluded from analysis. Unless otherwise noted, all survey items were developed by the present authors.

Participants first responded to items that probed their perceived risk from COVID-19 itself. Those items are shown in Table 1. The table displays the core question for each item: the exact wordings differed slightly between waves and scenarios and are available in the full survey texts. Those differences arose from inspection of the results from the first wave, and data gathered in parallel surveys in other countries, which were carefully examined to identify possible ambiguities or other problems with the items. All responses used a 5-point scale, where higher values always corresponded to endorsement of the issue being probed (e.g., 1 = Not at all

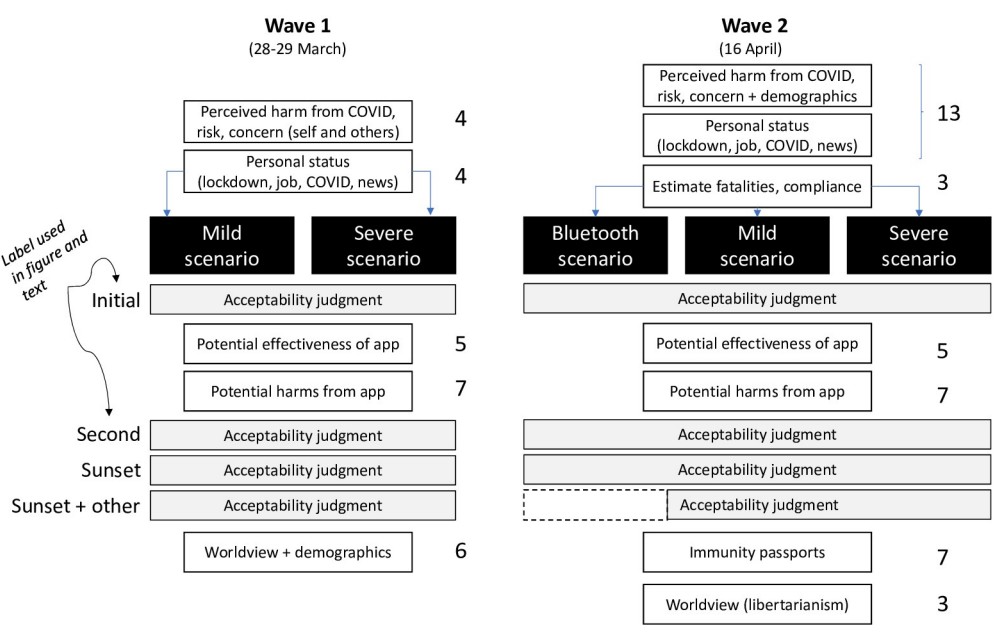

**Fig 1. Overview of surveys used in both waves.**

concerned to 5 = Extremely concerned). The labels and endpoints differed slightly between items and are available in the full survey texts. Participants were then randomly assigned to scenarios in each wave and were presented with the scenario text.

The text of the mild scenario was:

The COVID-19 pandemic has rapidly become a worldwide threat. Containing the virus' spread is essential to minimise the impact on the healthcare system, the economy, and save many lives. The U.K. Government might consider using smartphone tracking data to identify and contact those who may have been exposed to people with COVID-19. This would help reduce community spread by identifying those most at risk and allowing health services to be appropriately targeted. Only people who downloaded a government app and agreed to be tracked and contacted would be included in the project. The more people download and use this app, the more effectively the Government would be able to contain the spread of COVID-19. Data would be stored in an encrypted format on a secure server accessible only to the U.K. Government. Data would only be used to contact those who might have been exposed to COVID-19.

The severe scenario was:

The COVID-19 pandemic has rapidly become a worldwide threat. Containing the virus' spread is essential to minimise the impact on the healthcare system, the economy, and save

**Table 1. Items querying risks from COVID-19.**

| Question | Label |
|---|---|
| How severe do you think novel coronavirus (COVID-19) will be for the general population? | General harm |
| How harmful would it be for your health if you were to become infected COVID-19? | Personal harm |
| How concerned are you that you might become infected with COVID-19? | Concern self |
| How concerned are you that somebody you know might become infected with COVID-19? | Concern others |

many lives. The U.K. Government might consider using phone tracking data supplied by telecommunication companies to identify and contact those who may have been exposed to people with COVID-19. This would help reduce community spread by identifying those most at risk and allowing health services to be appropriately targeted. All people using a mobile phone would be included in the project, with no possibility to opt-out. Data would be stored in an encrypted format on a secure server accessible only to the U.K. Government which may use the data to locate people who were violating lockdown orders and enforce them with fines and arrests where necessary. Data would also be used to inform the appropriate public health response and to contact those who might have been exposed to COVID-19. Individual quarantine orders could be made on the basis of this data.

The Bluetooth scenario, used in the second wave only, was:

The COVID-19 pandemic has rapidly become a worldwide threat. Containing the virus' spread is essential to minimise the impact on the healthcare system, the economy, and save many lives. Apple and Google have proposed adding a contact tracing capability to existing smartphones to help inform people if they have been exposed to others with COVID-19. This would help reduce community spread of COVID-19 by allowing people to voluntarily self-isolate. When two people are near each other, their phones would connect via Bluetooth. If a person is later identified as being infected, the people they have been in close proximity to are then notified without the government knowing who they are. The use of this contact tracing capability would be completely voluntary. People who are notified would not be informed who had tested positive.

People's acceptability of the scenario was then probed 4 times (see left-most column in Fig 1). The initial test was immediately after presentation of the scenario. The wording of the acceptability question differed between scenarios to reflect the attributes of the policy. For the mild scenario, the question asked whether a participant "would download and use" the app, whereas for the severe scenario the question was whether the "use of tracking data in this scenario is acceptable." For the Bluetooth scenario, the participant was asked whether they "would use" the capability.

The second test occurred after a number of intervening questions that probed the perceived benefits of the tracking app described in the scenario, as well as the risks and harms that could arise from release of the personal data gathered in the process. Those items are shown in Table 2. The table again displays the core question for each item, with the exact wording available in the survey texts. All responses used a 6-point scale, where higher values always corresponded to endorsement of the issue being probed by the item (e.g., 1 = Not at all to 6 = Extremely). Items of differing polarity are identified by "[R]" in the table and were reverse scored before computing aggregate scores. For example, the items probing difficulty of declining participation and having control of their data are of different polarity because a consistent position would imply endorsement of one and rejection of the other. The scale labels and endpoints differed slightly between items and are available in the full survey texts.

The second test of acceptability was immediately followed by two more items that again queried acceptance of the scenario, but under modified assumptions. Those items were only presented to participants who found the scenario unacceptable on the second occasion. The first modification involved a sunset clause and queried whether deletion of the data after 6 months would make the scenario acceptable. The second modification differed betweeen scenarios and queried acceptability if the data were stored locally rather than on a government server (mild) or if users could opt out of data collection (severe). This second modified

**Table 2. Items querying potential benefits (Bfit) and harms (Harm) of app.**

| Item | Question | Label |
|---|---|---|
| Bfit 1 | How confident are you that the Government app would reduce your likelihood of contracting COVID-19? | Reduce contracting |
| Bfit 2 | How confident are you that the Government app would help you resume your normal activities more rapidly? | Resume normal |
| Bfit 3 | How confident are you that the Government app would reduce the spread of COVID-19? | Reduce spread |
| Harm 1 | How difficult is it for people to decline participation? | Difficult decline [R] |
| Harm 2 | To what extent do people have ongoing control of their data? | Have control |
| Harm 3 | How sensitive is the data being collected? | Sensitivity |
| Harm 4 | How serious is the risk of harm from the proposed policy? | Risk of tracking |
| Harm 5 | How secure is the data that would be collected? | Data security [R] |
| Harm 6 | To what extent is the Government only collecting the data necessary to achieve the purposes of the policy? | Proportionality |
| Harm 7 | How much do you trust the Government to use the tracking data only to deal with the COVID-19 pandemic? | Trust intentions |
| Harm 8 | How much do you trust the Government to be able to ensure the privacy of each individual? | Trust privacy |

assumption was not queried in the Bluetooth scenario (dotted empty field in the figure) because it already involved local storage and voluntary participation.

Assessment of acceptability was followed by an assessment of people's political worldviews, using three items that probed endorsement of free markets and small government. Two of those items were imported from previous research [21]; "An economic system based on free markets unrestrained by government interference automatically works best to meet human needs" and "The free market system may be efficient for resource allocation but it is limited in its capacity to promote social justice". The latter item was reverse coded for analysis. The third item ("The government should interfere with the lives of citizens as little as possible") was created for the purposes of this study. Worldview was scored such that higher average responses reflected more conservative-libertarian worldviews.

In the second wave, the worldview questions were preceded by a series of questions that probed people's attitudes to "immunity passports." Immunity passports were explained as follows:

An "immunity passport" indicates that you have had a disease and that you have the antibodies for the virus causing that disease. Having the antibodies implies that you are now immune and therefore unable to spread the virus to other people. Thus, if an antibody test indicates that you have had the disease, you could be allocated an immunity passport which would subsequently allow you to move around freely. Immunity passports have been proposed as a potential step towards lifting movement restrictions during the COVID-19 pandemic.

Table 3 explains the items used to query attitudes towards immunity passports. The table again displays the core question for each item, with the exact wording available in the survey text for wave 2. Responses used a 5-point or 6-point scale, where higher values always

**Table 3. Items querying attitudes towards immunity passports.**

| Question | Label | Scale |
|---|---|---|
| Would you support a government proposal to introduce immunity passports? | Support | 6 |
| How concerned are you about the idea of introducing an immunity passport? | Concern [R] | 5 |
| How much would you like to be allocated an immunity passport? | Like self | 6 |
| To what extent do you believe an immunity passport could harm the social fabric? | Harm general [R] | 6 |
| Is it fair for people with immunity passports to go back to work, while individuals without a passport cannot? | Fairness | 6 |
| To what extent would you consider infecting yourself with COVID-19 to get an immunity passport? | Infect self | 6 |
| Would you support a government proposal to introduce immunity passports? | Support 2 | 6 |

corresponded to endorsement of the issue being probed (e.g., 1 = Not at all to 6 = Extremely). The final column in the table indicates the scale (5 or 6 points) for each item. Items of different polarity that required reverse scoring before being aggregated into a composite score are identified by "[R]".

Altogether the surveys contained 32 (first wave) and 43 items (second wave), including one attention filter presented immediately after the scenario that tested participants' comprehension of the gist of each scenario. Because some questions were contingent on earlier responses, not all participants saw all items. Not all items in the surveys are analyzed and reported here although raw data for unreported items are made available at the link below.

Participants were invited to enter the survey through a link placed on Prolific. After reading an information sheet that explained the study, participants were given the option to provide consent and affirm that they were 18 or older via mouse click. Participants then responded to the items in the sequence shown in Fig 1. The survey concluded with debriefing information that included links to official websites with information about COVID-19 and resources for assistance for anxiety or other mental health concerns relating to the pandemic.

### Ethics statement

The study received ethics approval from the University of Bristol. Approval numbers 102663 (Wave 1) and 103344 (Wave 2). All participants gave informed consent and were fully debriefed. The full text of the information and debriefing sheets is included with the surveys at the links above.

## Results

### Data and source code availability

The data are available at https://osf.io/42wj6/ (first wave) and https://osf.io/pw5yj/ (second wave). Demographics and other sensitive variables (such as location information) that could lead to deanonymization have been omitted from the published data sets. The source code for analysis is embedded in the report at https://stephanlewandowsky.github.io/UKsocialLicence/index.html.

### Data preparation and demographics of sample

The requested number of participants was not obtainable in either wave in a reasonable time. The survey was discontinued after approximately 24 hours had elapsed since the last response was collected, which yielded a sample of $N = 1987$ and $N = 1493$ for the first and second wave,

**Table 4. Demographics for both waves.**

| | Gender | | | Age | |
|---|---|---|---|---|---|
| Wave | Male | Female | Other | Mean | SD |
| Wave 1 | 48.8% | 51% | 0.1% | 45.6 | 15.36 |
| Wave 2 | 48.2% | 51.7% | 0.1% | 46.15 | 15.32 |

respectively. After removal of duplicate responses from the same Prolific ID ($N = 142$ and $N = 1$, respectively, for wave 1 and 2), participants who failed the comprehension check, and incomplete responses, the final sample retained for analysis contained $N = 1810$ and $N = 1446$ for wave 1 and 2, respectively. The large number of duplicate responses in wave 1 arose from the need to run the survey in two batches on consecutive days because Prolific does not permit samples greater than 1,500. The second wave was run in a single batch but included a question whether a participant had taken the survey before. 174 respondents indicated yes, and a further 131 were unsure. Given the 3-week lag between waves, all those responses were retained for the second wave. Basic demographics are shown in Table 4 for both waves.

Self-reported education level is shown in Table 5. Samples from both waves were remarkably similar, although in both instances the share of respondents who indicated that they had a university education exceeded the official figure (latest release from Office of National Statistics in 2017 indicates 42% graduates in population; https://www.ons.gov.uk/employmentandlabourmarket/peopleinwork/employmentandemployeetypes/articles/graduatesintheuklabourmarket/2017). The difference may reflect the composition of the Prolific sample (which is stratified by age, sex and ethnicity but not education) or it may reflect a slight ambiguity in our question, which asked for the highest level of education completed, but did not specify successful graduation as a criterion. Anyone having attended university would therefore have been likely to choose "University education" from among the choices on offer.

## Perceived risk from COVID-19

Fig 2 shows the distribution of responses to the 4 items querying people's perceived risk from COVID-19 for both waves. The items are explained in Table 1. The figure shows that there were only small differences between the two waves. On both occasions, people expressed considerably more concern for others than for themselves.

## Attitudes towards tracking scenarios

The first part of the analysis focused on the various tracking policies sketched in the scenario presented to participants.

**Overall acceptability of scenarios.** Fig 3 shows acceptability ratings for the first wave of the survey, and Fig 4 shows the same ratings for the second wave. The four groups of bars in each figure refer to the 4 occasions on which acceptability was queried in the survey (see Fig 1). The last two items were only presented to participants who found the scenario unacceptable at the second test. The bars in the figure for those items represent the acceptable responses

**Table 5. Self-reported level of education for both waves.**

| | Education | | | |
|---|---|---|---|---|
| Wave | GCSE | A levels/VCE | University | Apprent/Vocatnl |
| Wave 1 | 15.3 | 17.3 | 55.6 | 11.7 |
| Wave 2 | 14.6 | 17.2 | 56.4 | NA |

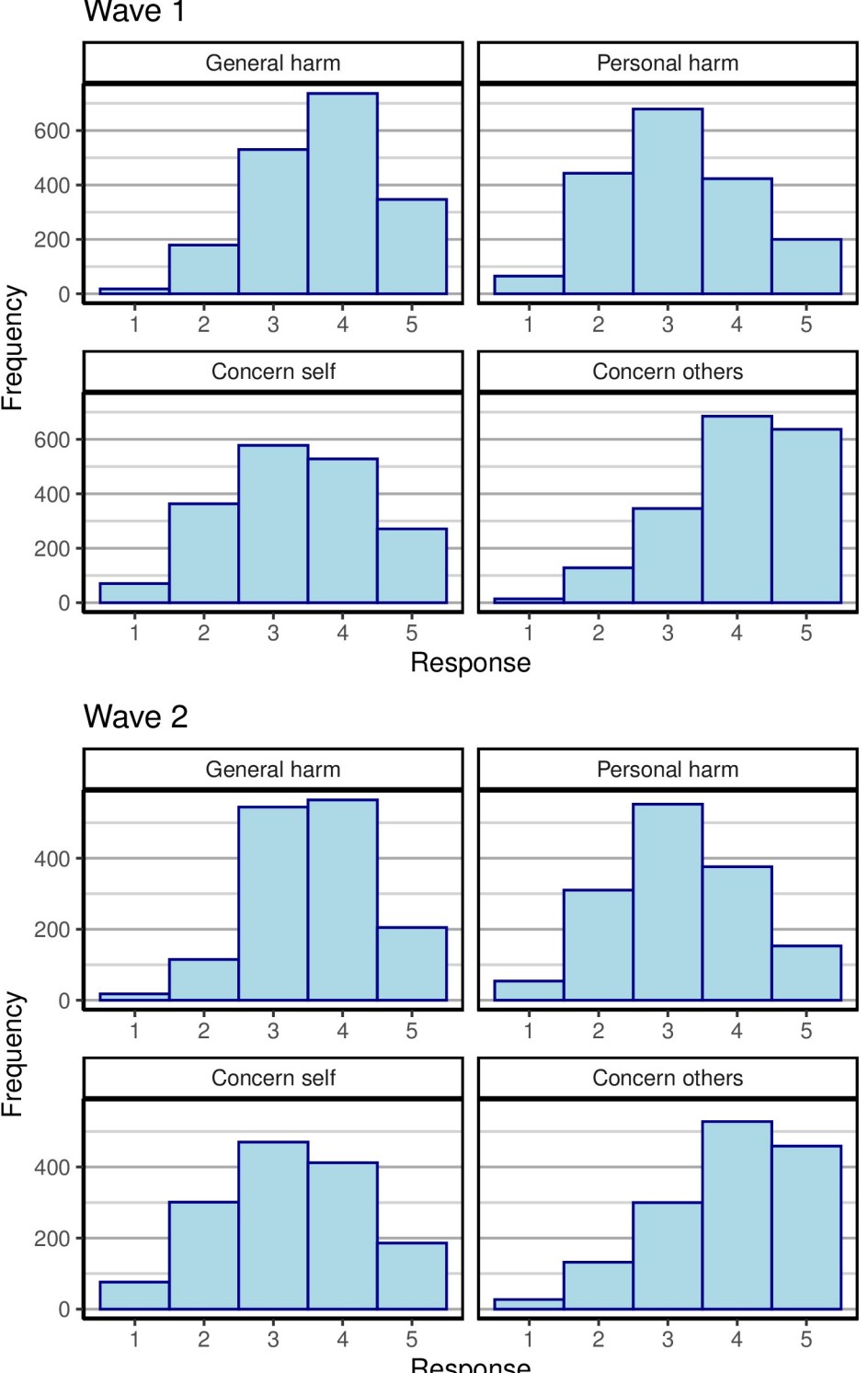

**Fig 2. Perceived risk from COVID during the first wave (28-29 March 2020; top set of panels) and during the second wave (16 April 2020; bottom panels).** Each panel plots responses for one item. See Table 1 for explanation of the items.

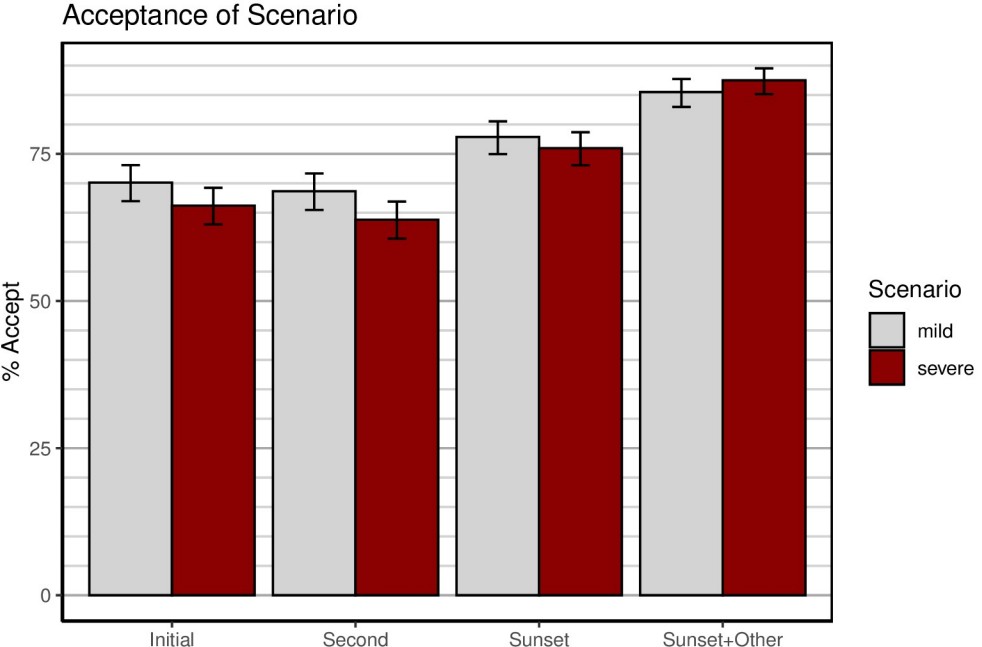

**Fig 3. Acceptability of scenarios during the first wave (28-29 March 2020).** The 4 pairs of bars refer to the 4 acceptability questions; see Fig 1. Error bars are 95% confidence intervals computed by R function *prop.test*.

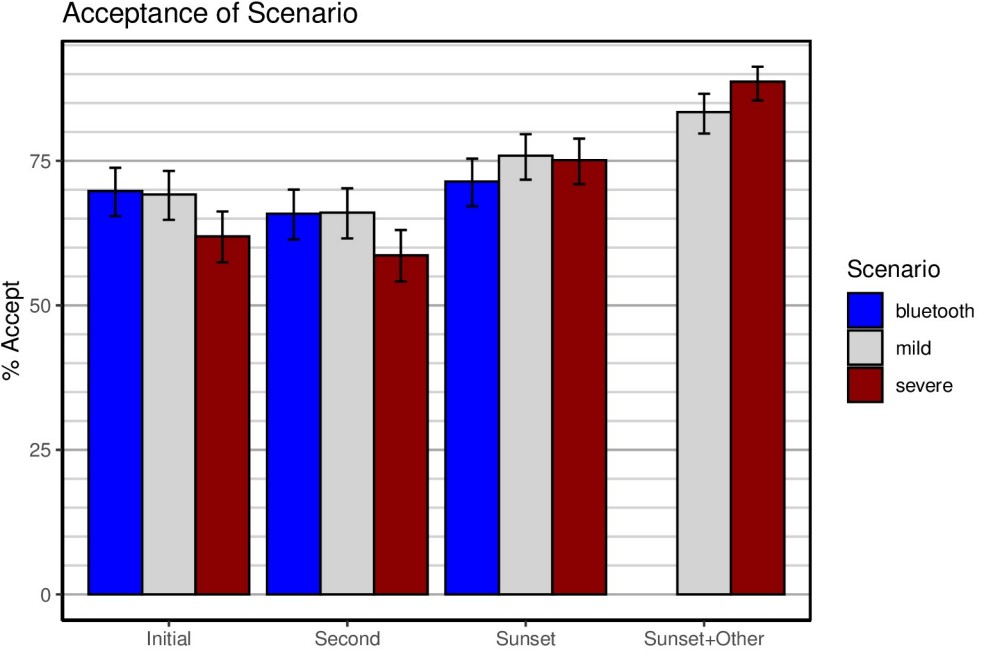

**Fig 4. Acceptability of scenarios during the second wave (16 April 2020).** The 4 pairs of bars refer to the 4 acceptability questions; see Fig 1. Error bars are 95% confidence intervals computed by R function *prop.test*. Note that the "Sunset + other" question was not presented in the Bluetooth scenario.

from the second test and the *additional* acceptance elicited by addition of a sunset clause and/
or local storage or opt-out.

Overall, acceptance of tracking technologies was quite high, with a baseline at the initial test
of around 70% for the mild and Bluetooth scenarios, and above 60% for the severe scenario.
To examine the effect of the different scenarios, a separate logistic regression was conducted
on data from each wave with the type of scenario entered as the only factor and the acceptance
response serving as binary dependent variable. For the first wave, the difference between sce-
narios in initial acceptance approached, but failed to reach, conventional significance levels,
$b = -0.18$, 95% CI [$-0.38$, $0.02$], $z = -1.79$, $p = .074$. For the second wave, the mild and the
Bluetooth scenarios did not differ from each other, $b = 0.03$, 95% CI [$-0.25$, $0.30$], $z = 0.20$, $p =
.843$, whereas the difference between the mild and the severe scenarios was significant, $b =
-0.32$, 95% CI [$-0.59$, $-0.06$], $z = -2.36$, $p = .018$.

Acceptance of all scenarios declined slightly at the second test, after participants responded
to more detailed questions about benefits and potential harms of the scenarios (reported next).
The addition of a sunset clause boosted acceptance for all scenarios, as did the provision of
local storage (for mild) and opt-out (severe). Indeed, with an opt-out clause, the severe sce-
nario achieved nearly 90% acceptance. It must be borne in mind, however, that the severe sce-
nario with an opt-out clause closely resembles the mild scenario, the remaining difference
being that the mild scenario requires opt-in rather than permitting opt-out.

**Potential effectiveness of tracking.** Fig 5 shows responses from both waves to three ques-
tions about the perceived benefits of the tracking scenario. The panels and Y-axes use the labels
from Table 2. It is clear that the severe scenario is judged to be slightly, but not dramatically,
more effective than the other two for at least some of the items. The two waves do not appear
to differ appreciably from each other.

Separate one-way ANOVAs for each wave and item confirm the pattern in the figure, with
an effect of scenario type on Reduce Contracting in wave 1, $F(1, 1808) = 5.60$, $MSE = 1.88$, $p =
.018$, $\hat{\eta}_G^2 = .003$, and in wave 2, $F(2, 1443) = 13.13$, $MSE = 1.75$, $p < .001$, $\hat{\eta}_G^2 = .018$. Follow-
up comparisons by Tukey HSD revealed that the severe scenario differed from each of the
others in wave 2, whereas the mild and Bluetooth scenarios did not differ from each other.
For Resume Normal, scenario type had no effect in wave 1, $F(1, 1808) = 0.84$, $MSE = 1.93$, $p =
.360$, $\hat{\eta}_G^2 = .000$, but it did have an effect in wave 2, $F(2, 1443) = 5.01$, $MSE = 1.81$, $p = .007$,
$\hat{\eta}_G^2 = .007$, which arose from a significant difference between the severe and Bluetooth scenar-
ios. No other pairwise tests were significant. Finally, for Reduce Spread, scenarios differed sig-
nificantly in wave 1, $F(1, 1808) = 5.63$, $MSE = 1.91$, $p = .018$, $\hat{\eta}_G^2 = .003$, and wave 2, $F(2, 1443)
= 8.06$, $MSE = 1.70$, $p < .001$, $\hat{\eta}_G^2 = .011$, with the difference in wave 2 arising from the severe
scenario leading to significantly higher confidence than the other two scenarios, which did not
differ from each other.

**Potential harms from tracking.** The items probing harm (Table 2) were aggregated into
three clusters that represented, respectively, people's perceived control over the policy (Items
Difficult decline [R] and Have control), harms from the policy (Sensitivity, Risk of tracking,
Data security [R]), and trust in government (Proportionality, Trust intentions, Trust privacy).
Items within each cluster were averaged after reverse scoring where appropriate (items identi-
fied with [R]).

Fig 6 shows responses from both waves to the three clusters of questions about the per-
ceived harms of the different tracking technologies. Each panel presents the composite score
for each cluster of items.

Separate one-way ANOVAs were conducted for each wave and item cluster where appro-
priate. For the control cluster, the severe scenario was omitted from any analysis because—

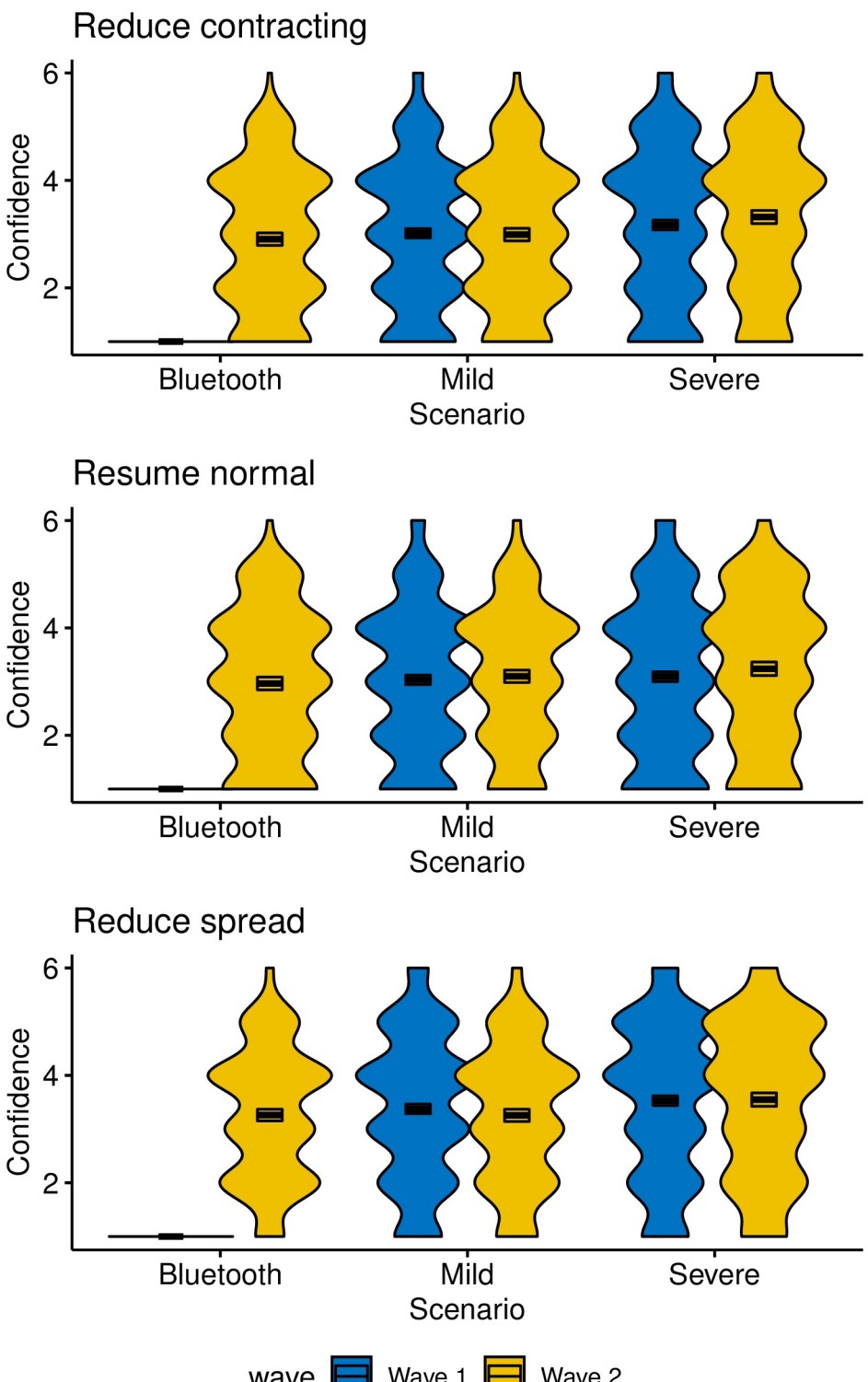

**Fig 5. Participants' confidence in the expected benefits from the three tacking policies.** Box plots enclose 95% confidence intervals. Panels represent different items, using the labeling from Table 2.

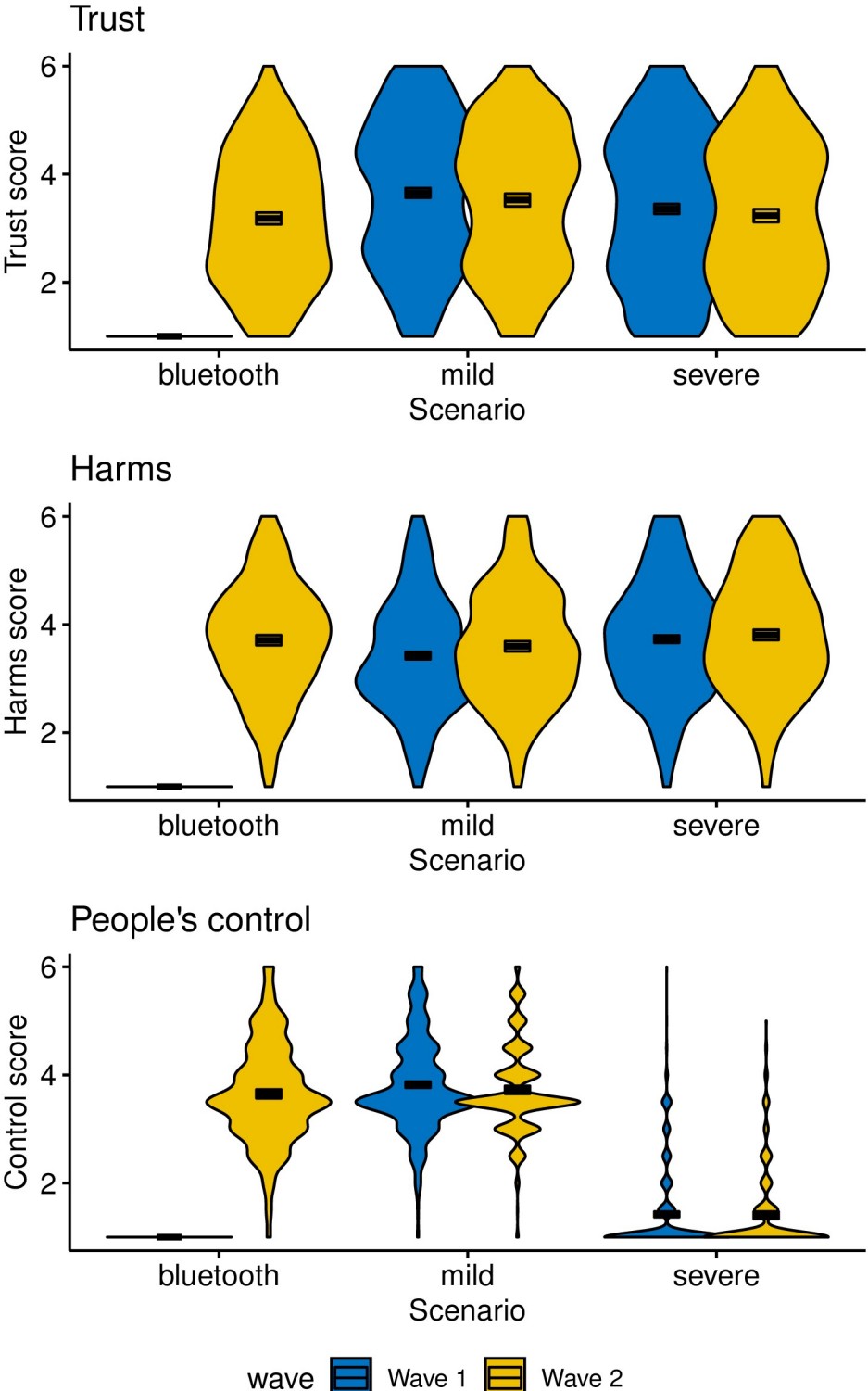

**Fig 6. Participants' views on the expected harms and risks of the three tacking policies and perceived trust in the government's intentions.** Box plots enclose 95% confidence intervals. Panels represent different item clusters; see text for details.

quite appropriately—most people recognized that it offered no control at all. The only ANOVA for the control cluster thus involved a comparison between the mild scenario and Bluetooth in wave 2, $F(1, 958) = 1.80$, $MSE = 0.76$, $p = .180$, $\hat{\eta}_G^2 = .002$, which revealed that the two scenarios did not differ from each other. For harms, differences between scenarios were significant for wave 1, $F(1, 1808) = 37.49$, $MSE = 1.12$, $p < .001$, $\hat{\eta}_G^2 = .020$, as well as for wave 2, $F(1, 1808) = 37.49$, $MSE = 1.12$, $p < .001$, $\hat{\eta}_G^2 = .020$, with the latter effect being exclusively driven by a difference between the severe and mild scenarios. For trust, effects were observed in wave 1, $F(1, 1808) = 20.96$, $MSE = 1.90$, $p < .001$, $\hat{\eta}_G^2 = .011$, and wave 2, $F(2, 1443) = 9.62$, $MSE = 1.68$, $p < .001$, $\hat{\eta}_G^2 = .013$, with the latter effect being driven by significant differences between the mild scenario and each of the other two.

**Predictors of tracking policy acceptance.** The final analysis modeled acceptance of the various policy scenarios as a function of several sets of predictors using logistic regression. The predictors included wave (first or second), demographics (age and gender, excluding respondents who did not choose "male" or "female"; $N = 6$), a measure of worldview aggregated across the three relevant items; perceived risk from COVID (aggregated score across items in Table 1); and the earlier aggregate scores for perceived harm from the policy and trust in government (top two panels of Fig 6; the bottom panel relating to control was omitted because there was no meaningful variance for the severe scenario). The initial acceptance of the scenario (i.e., the first time acceptance was probed) was used as the binary dependent variable.

We first attempted to fit a number of random effects models (e.g., with a different intercept for each participant) using the *lmer* function in R. All of these models failed to converge. The likely reason for this failure was near-zero variance of the random effect. We therefore fit a conventional logistic regression using *glm* in R with fixed effects only.

We compared two fixed-effect models: a complex model which included the above predictors and their interactions with wave, and a simpler model without any interaction terms. Removing the interactions incurred no significant loss of fit, $\chi^2(6) = 0.414$, p > .10. Moreover, the simpler model was preferred by BIC (2796 vs. 2845). We therefore only report the simpler model with wave functioning as a predictor but not interacting with any of the others. Fig 7 shows the estimated standardized regression coefficients for the final model (intercept omitted from figure). The model accounted for a substantial share of the variance, McFadden's pseudo-$r^2 = 0.33$ and Cragg and Uhler's pseudo-$r^2 = 0.48$.

The figure shows that increasing age was associated with reduced acceptance of a policy, and that men were less accepting than women overall. In addition, worldview had a small but consistent effect on policy acceptance, such that people with a more conservative or libertarian orientation were slightly less likely to accept any of the policies. Reduced acceptance was also associated with greater perceived harms from the tracking technologies. Conversely, greater perceived risk from COVID was associated with greater policy acceptance. Finally, by far the most important predictor turned out to be trust in government. People who trusted the government to safeguard privacy were considerably more likely to accept the policies than people who distrusted the government.

## Immunity passports

The second part of the analysis focused on attitudes towards immunity passports. This analysis is confined to wave 2 because questions about passports were limited to that wave.

**Acceptance of immunity passports.** Fig 8 displays the distribution of responses to all items querying immunity passports. The majority of people clearly did not object to the idea of passports, with concern being low on average and more than 60% of people wanting one for themselves to varying extents. There were, however, around 20% of respondents who

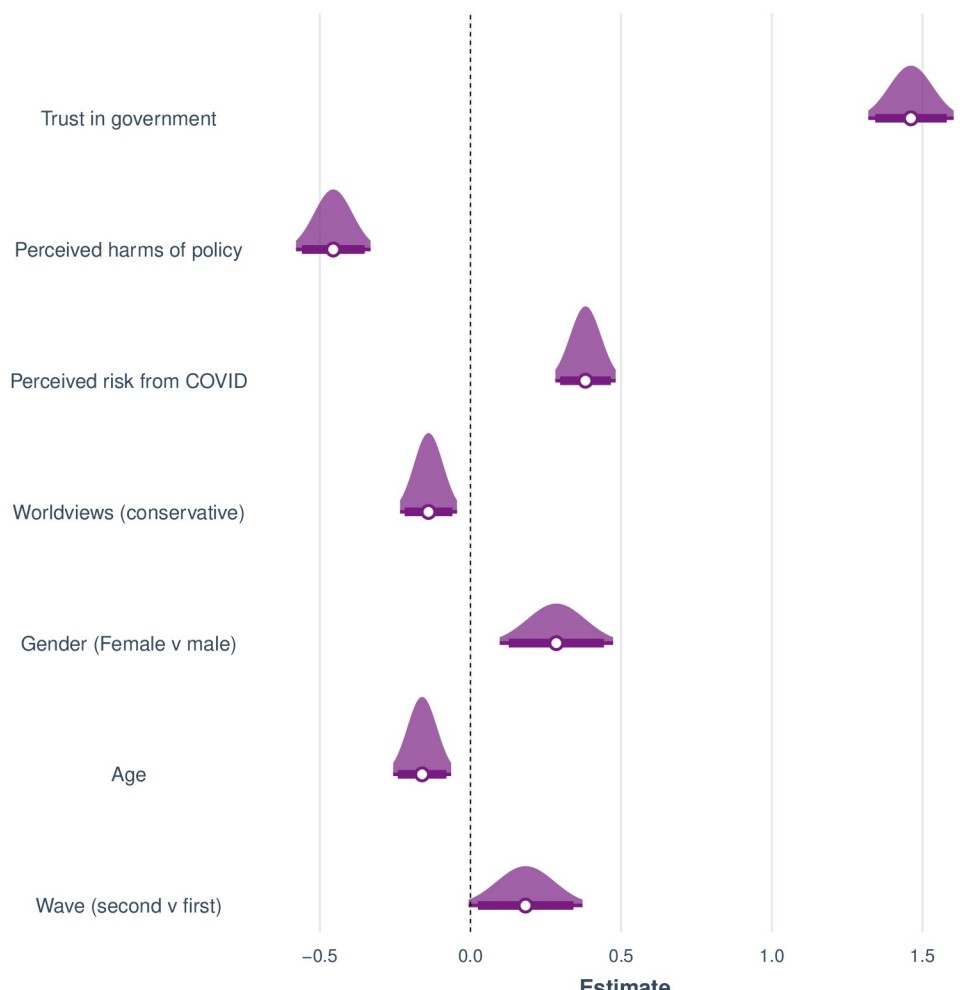

**Fig 7. Estimated standardized coefficients for a logistic regression to predict initial policy acceptance of the tracking scenarios.** Distributions span 95% confidence intervals. Horizontal bars span 90% confidence intervals.

considered passports to be unfair and who opposed them completely (response category 1 for the second support item).

**Predictors of immunity passport acceptance.** To model acceptance of immunity passports we created a composite score for immunity passports by averaging across all items in Table 3, reverse scoring where necessary. The "Infect self" item was excluded because it exhibited little variance and also did not correlate appreciably with most of the other items. The composite score was used as the dependent variable in a linear regression model that included most of the predictors from before; namely, age, gender (again excluding responses other than "male" or "female"), worldviews, perceived risk from COVID-19, and the perceived harms of the tracking policy and trust in government relating to that policy. Note that the latter two measures were gathered in connection with the tracking policy rather than immunity passports.

Fig 9 shows the regression coefficients for this model. The model accounted for a moderate share of variance, $r^2 = 0.17$, adjusted $r^2 = 0.17$. Most of the variance accounted for was due to the two predictors relating to the tracking scenario (Perceived harms and Trust in government). When they were removed from the model, the explained variance was small, $r^2 = 0.03$,

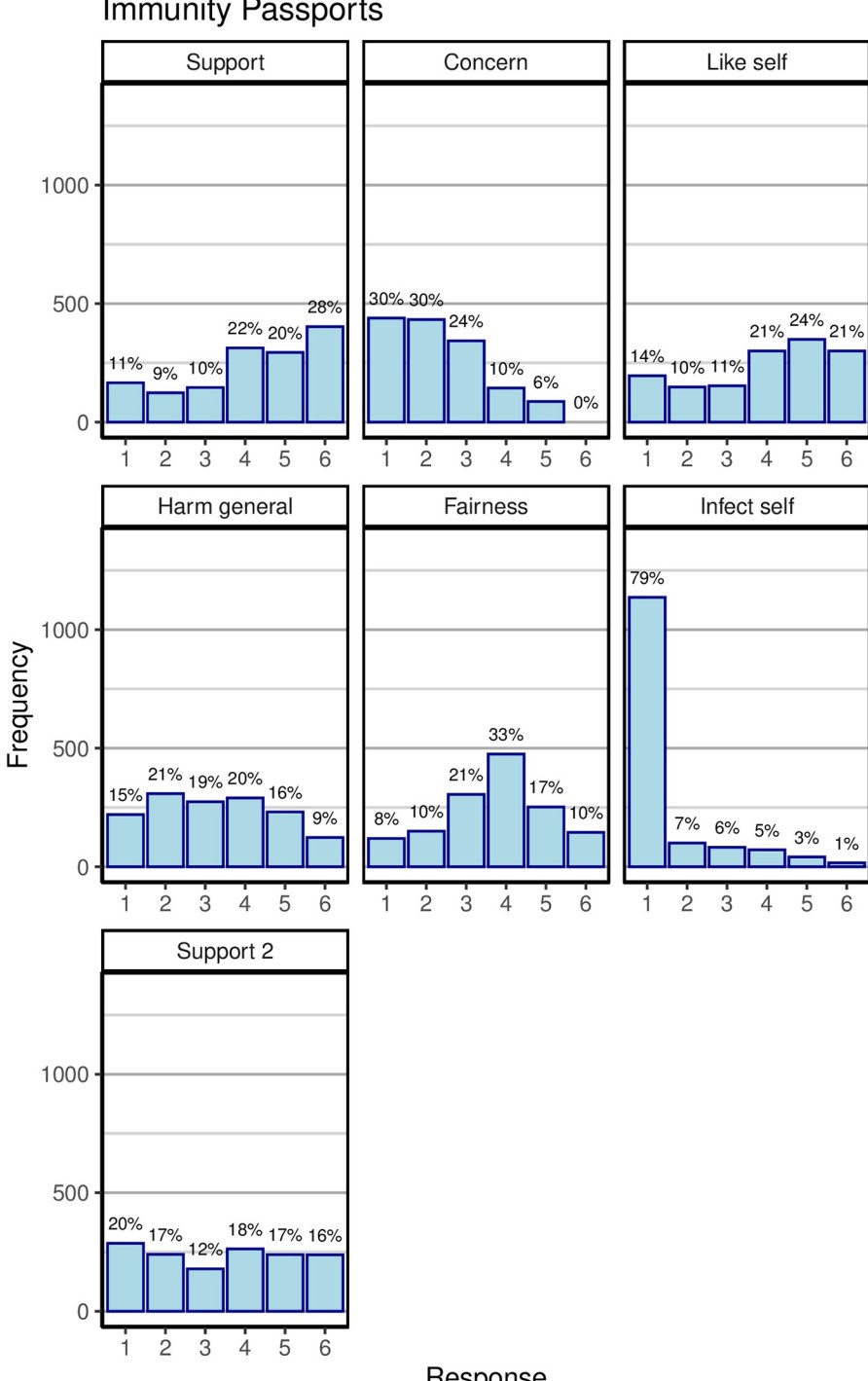

**Fig 8. Responses to items concerning immunity passports during wave 2.** Each panel plots responses for one item before reverse scoring. See Table 3 for explanation of the items. The "concern" item used a 5-point scale.

adjusted $r^2$ = 0.02, although age retained its significant effect, $t(1439) = 4.16$, $p < .001$, and the effect of perceived risk from COVID-19 was now highly significant, $t(1439) = 3.31$, $p = .001$.

In contrast to the tracking policies, increasing age was associated with increased acceptance of immunity passports whereas gender had no effect here. The other predictors relating to

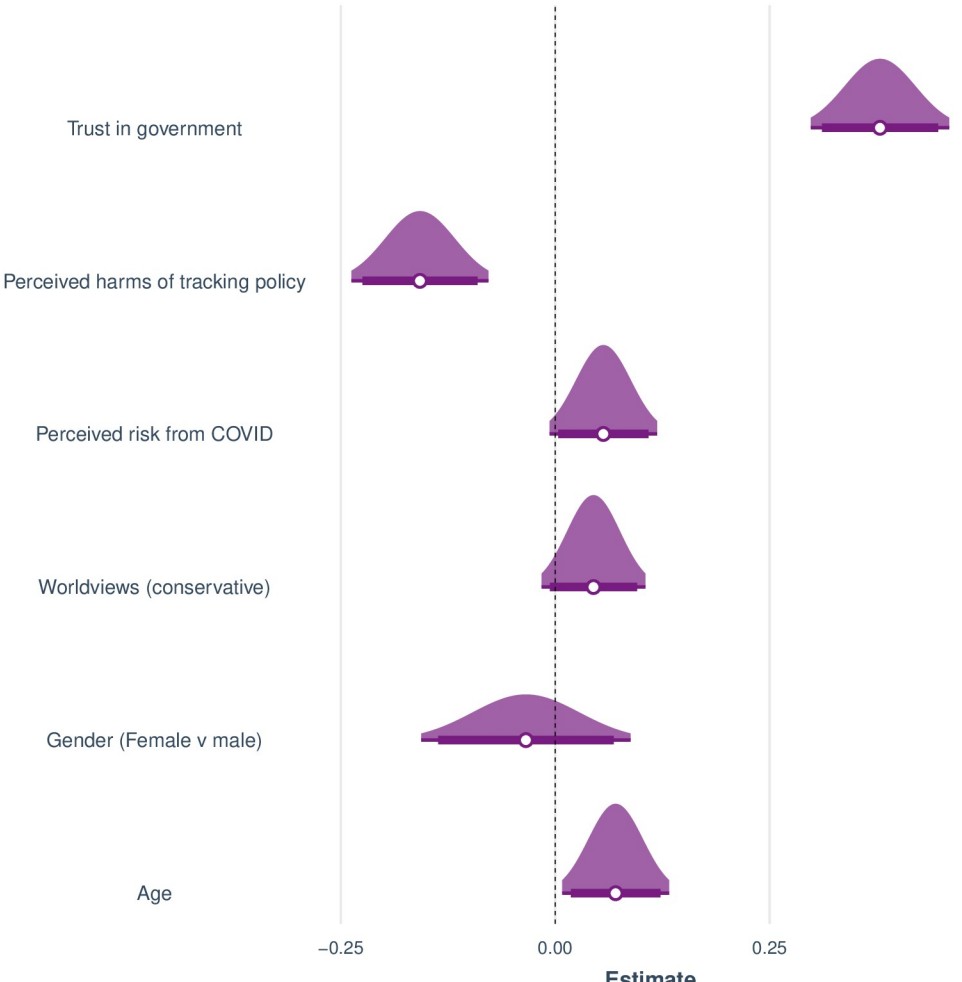

**Fig 9. Estimated standardized coefficients for a linear regression to predict favourable attitudes towards immunity passports.** Distributions span 95% confidence intervals. Horizontal bars span 90% confidence intervals. Note that Trust in government and Perceived harms are obtained in response to the tracking policy scenario, not immunity passports.

perceived risks predicted attitudes towards immunity passports as one might expect: greater perceived risk of the disease and greater trust in government were associated with more favorable attitudes whereas skepticism towards the tracking policies was also associated with greater skepticism towards immunity passports.

## Discussion

### Potential limitations

Our surveys only included three different tracking scenarios, which is a small number relative to the total number of technological solutions now available [22]. We also presented scenarios that differed somewhat from those currently in use around the world. This was unavoidable given that not many mature technologies existed at the time of our survey. Nonetheless, the essence of the Bluetooth scenario described in wave 2 has now become a reality in many countries, including the United Kingdom.

In light of the relatively slight differences in acceptance between scenarios, it might be argued that the participants did not fully understand the implications of the different hypothetical tracking apps. We do not believe that this argument has much force: As shown in Fig 6 (bottom panel), people are very aware of how much control they have over their data in the different scenarios. Virtually all participants recognized, for example, that the severe scenario offers no control over people's personal data. In contrast, participants correctly recognized greater control in the mild and Bluetooth scenarios. These results are not easily reconciled with the proposition that participants failed to understand the scenarios.

## Relationship to previous results

Our results mesh extremely well with those reported by an independent team of authors in the U.K. [23]. Similar to both our survey waves, [23] found that up to 75% of respondents would install the app. One difference between the studies is that we specified data storage and access precisely, whereas [23] did not explicitly explain where the data would be held (i.e., locally without location information or centrally). Their scenario was, however, closest to our Bluetooth scenario, which renders the two sets of acceptance probabilities nearly identical. The present data are also consonant with recent results from Poland [24]. People in Poland, polled in late March 2020, were also sensitive to the risk from COVID-19, with greater perceived risk being associated with greater endorsement of tracking technologies. The role of political views was more nuanced: Wnuk et al. found that people higher in right-wing authoritarianism were more likely to endorse tracking technologies, whereas endorsement of liberty (the single item "The freedom to do what we want is more important than following the recommendations of the authorities") was negatively associated with acceptance of tracking technologies.

Our results are somewhat more at variance with another recent report on app acceptance in the U.K. [25]. That study recruited a very large sample ($N \simeq 12{,}500$) in mid-May 2020 through the Care Information Exchange (CIE), a patient-facing NHS web platform. Around 60% of participants indicated willingness to download a contact-tracing app, although the study did not elaborate on the app other than to describe it as "an NHS app for your phone (like the one being tested on the Isle of Wight)." The majority of participants who declined participation (67% of that group) indicated that their refusal was due to privacy concerns. There are several possible reasons for the greater rates of endorsement observed here compared to [25]. In addition to obvious differences between samples (representative vs. self-selected users of an NHS website), we explained our scenarios in greater depth, and the additional information may have swayed people because it explained the public-health benefit of the technology.

There are also similarities between the two sets of results. Similar to us, [25] found that advanced age (i.e., 80 and above) was associated with decreased likelihood of participation. In addition, a reduced understanding of government rules and advice on the lockdown at the time was associated with reduced willingness to download the app. Moreover, people who thought they had COVID-19 and had recovered from it (without however being tested at any point), were 27% less likely to download the app. This underscores the important role played by variables that, at first glance, appear extraneous to a decision about app, but create the critically-important context for people's acceptance of tracking technologies.

We are not aware of any existing data on people's acceptance of immunity passports.

## Implications for policy

Our results have clear implications for policy. Perhaps the most important finding is the high overall level of endorsement for both policy options: A majority of people supports immunity passports, and an even greater majority endorses tracking-based policies. This high level of

endorsement stands in contrast to people's commonly professed concern for their privacy [26]. It appears that the British public is prepared to sacrifice some privacy in the interest of public health, in particular when the scenarios are explained in detail (compare our results to those of [25]).

A second implication of our results is that the fundamentals of the policy mattered relatively little: In both waves, the initial acceptance of the mild scenario was only 5%—10% greater than acceptance of the draconian severe scenario. This small difference is surprising in light of people's responses to opinion surveys which place a high value on privacy [27]. The small difference is, however, consonant with the fact that people tend to reveal personal information for relatively small rewards, contrary to their stated opinion [26–28]. Participants were nonetheless quite sensitive to other aspects of the scenarios, as revealed by the relatively large effect of the addition of a sunset clause or other measures such as local storage of the data.

Our results also reveal people to engage in a readily-understandable privacy calculus. Specifically, people trade off the perceived harms from the policy under consideration (tracking apps or immunity passports) against the perceived risk from COVID-19: increased risk perception increases policy acceptance and increased fear of fallout from the policies reduces support. The magnitude of the opposing effects of those two variables was roughly equal for tracking technologies, whereas for immunity passports, perceived harm outweighed perceived risk of COVID-19. Similar tradeoffs between opposing risks and benefits have been observed previously in the context of other mobile applications [29]. The most important driver of acceptance of both policies was a variable unrelated to perceptions of immediate risks and harms; namely, people's trust in the government's intention or ability to secure people's privacy and to manage access to the data.

A further important aspect of our result is that neither tracking technologies nor immunity passports appeared to be highly politicized, at least at the time the surveys were conducted. Although people with a conservative-libertarian worldview were less likely to accept tracking technologies (while being slightly more likely to endorse immunity passports), the effect sizes were modest relative to the other variables that were observed to enter into the privacy calculus here.

These implications can be combined into a straightforward policy for tracking apps: People are relatively unconcerned about where the data are stored, but they do care about how long the data will be stored for. Any policy should therefore be accompanied by a clear sunset clause, and that sunset clause should be highlighted in communications with the public. In addition, given the important role of trust, any policy rollout should be accompanied by clear messages about why and how the government is a trustworthy custodian of people's data. Those messages must also be easy to understand and follow [25]. This may require the government to explicitly step aside and highlight the fact that the data will be held by trusted institutions, such as the NHS.

The implications for immunity passports are somewhat more nuanced. Although there was a large degree of support for passports overall, one fifth of our sample strictly opposed their introduction. It is an open question whether this opposition would create an insurmountable political challenge to any government trying to introduce immunity passports.

## Author Contributions

**Conceptualization:** Stephan Lewandowsky, Simon Dennis, Andrew Perfors, Yoshihisa Kashima, Daniel R. Little.

**Data curation:** Stephan Lewandowsky.

**Formal analysis:** Stephan Lewandowsky, Paul Garrett, Daniel R. Little.

**Funding acquisition:** Stephan Lewandowsky.

**Methodology:** Stephan Lewandowsky, Simon Dennis, Joshua P. White, Daniel R. Little, Muhsin Yesilada.

**Project administration:** Stephan Lewandowsky.

**Software:** Joshua P. White.

**Visualization:** Stephan Lewandowsky, Paul Garrett.

**Writing – original draft:** Stephan Lewandowsky.

**Writing – review & editing:** Stephan Lewandowsky, Simon Dennis, Andrew Perfors, Yoshihisa Kashima, Paul Garrett, Daniel R. Little.

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
