## [Decision Letter · Decision Letter 0]

16 Oct 2020

PONE-D-20-26680

Public acceptance of Privacy-Encroaching Policies to Address the COVID-19 Pandemic in the United Kingdom

PLOS ONE

Dear Prof Lewandowsky,

Thank you for submitting your manuscript to PLOS ONE. After careful consideration, we feel that it has merit but does not fully meet PLOS ONE’s publication criteria as it currently stands. Therefore, we invite you to submit a revised version of the manuscript that addresses the points raised during the review process.

Please consider the reviewers comments. Please check the additional 

We look forward to receiving your revised manuscript.

Kind regards,

Andrew Soundy

Academic Editor

PLOS ONE

Journal Requirements:

Reviewers' comments:

Reviewer's Responses to Questions

**Comments to the Author**

1. Is the manuscript technically sound, and do the data support the conclusions?

Reviewer #1: Yes

Reviewer #2: Partly

2. Has the statistical analysis been performed appropriately and rigorously? 

Reviewer #1: Yes

Reviewer #2: Yes

3. Have the authors made all data underlying the findings in their manuscript fully available?

Reviewer #1: Yes

Reviewer #2: Yes

4. Is the manuscript presented in an intelligible fashion and written in standard English?

Reviewer #1: Yes

Reviewer #2: Yes

5. Review Comments to the Author

Reviewer #1: There are items for further explanations or clarifications:

1. The first survey was conducted on 28 and 29 March, and the second on 16 April. This means that the interval between the two is less than 3 weeks. If there is a reason for this short interval, it should be explained. Between the two surveys, perhaps the only significant difference explained in the article is the number of those who were infected (and died). If so, it would have been easily expected that the results from the two surveys would not be significantly different. The actual results indeed are not very different between the two.

2. Statistical discrepancies depending on scenarios (Mild, Severe, and Bluetooth) are not very large. One possible reason for this could be the lack of understanding about the differences among the scenarios. For instance, the Mild scenario is presumably a centralized scheme, whereas the Bluetooth scenario is a de-centralized scheme. Survey participants may or may not have fully appreciated the nuanced social and technological differences associated with each of the scenarios. The extent to which the survey participants may or may not have understood these differences need to be explained more clearly.

Reviewer #2: In the paper “Public acceptance of Privacy-Encroaching Policies to Address the COVID-19 Pandemic in the United Kingdom,” the authors conducted two surveys with participants from the United Kingdom. The first survey was conducted during the first wave of COVID-19, and the second survey during the second wave. Results are presented and discussed.

I think it’s important that you explain the privacy paradox and privacy calculus early in the paper, which have become important aspects in privacy research. However, later this is only discussed at a surface level. Please discuss much more in-depth. What are the implications for the privacy paradox and privacy calculus?

Please outline early in the paper what the main objective and contribution of the paper is, both for research and practice. In the last paragraph of the section “Immunity passports” you describe what you are doing in the paper but the objective and contribution is not clear. Please move this paragraph to the end of the introduction.

The method is described well, however, you could improve on transparency. In the section “Overview” you mention that an app was used. Did you develop the app? Is it a third-party app? Which app is it actually? Please provide links and screenshots. Did participants actually used the app?

Please describe in detail how the items were developed. Are they self-developed or did you draw on items from other prior studies? If self-developed, describe in detail the process of developing the items.

The results are interesting but it’s not clear how it will help for the current and any future pandemics. Is the main message that governments should build trust in government and reduce perceived harms of tracking policy? It appears to me that this is a too simplistic approach.

Important literature is missing. Please see the following review articles for reference to further literature:

Belanger, F., Crossler, R. E. 2011. Privacy in the Digital Age: A Review of Information Privacy Research in Information Systems. MIS Quarterly, 35(4), pp. 1017-1041. https://doi.org/10.2307/41409971

Smith, H. J., Dinev, T., Xu, H. 2011. Information Privacy Research: An Interdisciplinary Review. MIS Quarterly, 35(4), pp. 989-1015. https://doi.org/10.2307/41409970

You might also want to have a closer look at privacy literature with a specific mobile app context, see for example:

Degirmenci, K. 2020. Mobile users’ information privacy concerns and the role of app permission requests. International Journal of Information Management, 50, pp. 261-272. https://doi.org/10.1016/j.ijinfomgt.2019.05.010

Gu, J., Xu, Y., Xu, H., Zhang, C., Ling, H. 2017. Privacy concerns for mobile app download: An elaboration likelihood model perspective. Decision Support Systems, 94, pp. 19-28. https://doi.org/10.1016/j.dss.2016.10.002

Wang, T., Duong, T. D., Chen, C. C. 2016. Intention to disclose personal information via mobile applications: A privacy calculus perspective. International Journal of Information Management, 36, pp. 531-542. https://doi.org/10.1016/j.ijinfomgt.2016.03.003

Xu, H., Teo, H.-H., Tan, B. C. Y., Agarwal, R. 2009. The Role of Push-Pull Technology in Privacy Calculus: The Case of Location-Based Services. Journal of Management Information Systems, 26(3), pp. 135-173. https://doi.org/10.2753/MIS0742-1222260305

Xu, H., Teo, H.-H., Tan, B. C. Y., Agarwal, R. 2012. Effects of Individual Self-Protection, Industry Self-Regulation, and Government Regulation on Privacy Concerns: A Study of Location-Based Services. Information Systems Research, 23(4), pp. 1342-1363. https://doi.org/10.1287/isre.1120.0416

How does your paper extend the current body of knowledge?

Minor issues:

- There are issues with the structure. Please start with an introduction and summarise the sections “Tracking technologies” and “Immunity passports” under, e.g., “Background”.

- Change “sonsiderable” to “considerable” on line 77 on p. 4

6. PLOS authors have the option to publish the peer review history of their article (what does this mean?). If published, this will include your full peer review and any attached files.

Reviewer #1: No

Reviewer #2: No

---

## [Editor Report · Decision Letter 1]

7 Jan 2021

Public acceptance of Privacy-Encroaching Policies to Address the COVID-19 Pandemic in the United Kingdom

PONE-D-20-26680R1

Dear Dr. Lewandowsky,

We’re pleased to inform you that your manuscript has been judged scientifically suitable for publication and will be formally accepted for publication once it meets all outstanding technical requirements.

Kind regards,

Andrew Soundy

Academic Editor

PLOS ONE
---

## [Editor Report · Acceptance letter]

11 Jan 2021

PONE-D-20-26680R1 

Public acceptance of Privacy-Encroaching Policies to Address the COVID-19 Pandemic in the United Kingdom 

Dear Dr. Lewandowsky:

I'm pleased to inform you that your manuscript has been deemed suitable for publication in PLOS ONE. Congratulations! Your manuscript is now with our production department. 

Kind regards, 

on behalf of

Dr. Andrew Soundy 

Academic Editor

PLOS ONE